# Adult Diabetes and Prediabetes Prevalence in Kuwait: Data from the Cross-Sectional Kuwait Diabetes Epidemiology Program

**DOI:** 10.3390/jcm9113420

**Published:** 2020-10-25

**Authors:** Abdullah Alkandari, Monira Alarouj, Naser Elkum, Prem Sharma, Sriraman Devarajan, Mohamed Abu-Farha, Fahd Al-Mulla, Jaakko Tuomilehto, Abdullah Bennakhi

**Affiliations:** 1Dasman Diabetes Institute, Kuwait City 15462, Kuwait; monira.arouj@dasmaninstitute.org (M.A.); prem.sharma@dasmaninstitute.org (P.S.); sriraman.devarajan@dasmaninstitute.org (S.D.); mohamed.abufarha@dasmaninstitute.org (M.A.-F.); fahd.almulla@dasmaninstitute.org (F.A.-M.); abdullah.bennakhi@dasmaninstitute.org (A.B.); 2Sidra Medical and Research Center, Doha 26999, Qatar; nelkum@sidra.org; 3Public Health Prevention Unit, Finnish Institute for Health and Welfare, FI-00271 Helsinki, Finland; jaakko.tuomilehto@thl.fi

**Keywords:** diabetes, prediabetes, prevalence, Kuwait, epidemiology

## Abstract

Background: This study aimed to estimate the prevalence of diabetes and prediabetes in adults in Kuwait. Methods: The Kuwait Diabetes Epidemiology Program was a nationally representative, cross-sectional study of diabetes and obesity in Kuwait conducted between 2011 and 2014. The survey sampled 4937 adults in Kuwait aged 20 years or more and recorded participants’ demographics, behaviours, medical history, physical measurements and blood biochemical measurements. Prediabetes was defined as fasting plasma glucose between 6.1 and 6.9 mmol/L or HbA1c between 6 and 6.4% (42–47 mmol/mol). Diabetes was defined as self-reported history with prescribed glucose-lowering medication or FPG ≥7mmol/L or HbA1c level ≥6.5% (≥48 mmol/mol). Results: The overall adjusted prevalence of diabetes was 19.1%. The overall adjusted prevalence of prediabetes was 13.5%. Diabetes prevalence was 5.4%, 14.2%, 38.7% and 64.8% in adults aged 20–29, 30–44, 45–59 and 60 years or more, respectively. Diabetes prevalence was 22.4% in men and 14.4% in women. Prediabetes prevalence was 14.8% in men and 11.5% in women. In Kuwaitis, diabetes and prediabetes prevalence was 21.8% and 11.1%, respectively, while prevalence in non-Kuwaitis was 18.2% for diabetes and 14.3% for prediabetes. Conclusion: These findings illustrate the severe public health challenge posed by diabetes in Kuwait.

## 1. Introduction

The global prevalence of diabetes continues to rise. The prevalence of diabetes was estimated by the International Diabetes Federation (IDF) to be 9.3% in 2019, increased from 4.6% in 2000 in adults aged 20–79 years [1]. The age-adjusted prevalence of adult diabetes in the Middle East and North Africa (MENA) region, which includes Kuwait, was 12.2%, the highest estimated prevalence of all the IDF regions. Prevalence in the MENA region is expected to increase to 13.9% by 2045 [1].

Kuwait is a small country nestled between Iraq and Saudi Arabia on the Arabian Peninsula. The discovery of oil has transformed Kuwait into a wealthy country with a largely expatriate workforce, over two-thirds of which are men [2]. This economic transformation has led to rapid urbanisation, a sedentary lifestyle and lack of physical exercise, which has in turn led to a rise in noncommunicable diseases (NCDs). A recent World Health Organization (WHO) STEPS cross-sectional survey found that almost 80% of Kuwaiti adults were overweight or obese in 2014 [3], and almost 40% had diabetes or prediabetes [4]. The socioeconomic burden of this mounting public health challenge has been large; Kuwait spends over $1000 (USD) per adult with diabetes [1], and NCDs were estimated to be the cause of 72% of all deaths in Kuwait in 2016 [5].

The Kuwait Diabetes Epidemiology Program (KDEP) was a national, cross-sectional, population-based study of noncommunicable diseases. Unlike the 2014 WHO STEPS survey [4], KDEP sampled both Kuwaiti and non-Kuwaiti adults. We have previously reported cardiovascular disease risk factors, including the prevalence of impaired fasting glucose and diabetes, in the South Asian population of KDEP [6]. Here, we describe the prevalence of diabetes and prediabetes in the entire surveyed population.

## 2. Experimental Section

### 2.1. Participants and Study Design

KDEP was a cross-sectional, population-based survey on noncommunicable diseases conducted on a representative sample of adults in Kuwait (aged ≥18 years) between 2011 and 2014. A detailed methodology has been described previously [6]. Briefly, a simple random sample with proportional allocation of all six administrative regions of Kuwait was provided by the national Public Authority of Civil Information based on the unique identification code assigned to all residents in Kuwait. A stratified random sampling technique was used to select the survey participants. Interviews were completed for 5291 individuals. The overall participation rate was 85.1%.

The survey design was adapted from the WHO STEPwise approach to surveillance methodology [7] and comprised four consecutive steps. Step 1 consisted of a structured demographic questionnaire and step 2 recorded behavioural measures, including diet and lifestyle as well as participants’ medical history and family medical history (parents and siblings). Step 3 obtained physical measurements, including height, weight, waist circumference and blood pressure. Finally, step 4 consisted of a blood biochemistry analysis, which included fasting plasma glucose (FPG) and glycated haemoglobin (HbA1c) measurements. A total of 5291 adults participated in the first three steps and 4947 adults completed all four steps, including providing an FPG and an HbA1c sample.

### 2.2. Data Collection

Data collection occurred between April 2011 and June 2014 at the Dasman Diabetes Institute and was conducted by a multilingual team of nurses, study coordinators, interviewers, phlebotomists and supervisors. Recruited by phone, participants were instructed to visit the institute following a 10 h fast. The study was described to all participants on arrival and a written informed consent was obtained from each participant prior to their enrolment. Interviews were conducted in English or Arabic.

Height was measured using portable inflexible bars and weight was measured using calibrated portable electronic weighing scales. BMI was calculated as weight (kilograms) divided by height (metres) squared. Waist and hip circumferences were measured with a constant tension tape. Blood pressure was measured with an Omron HEM-907XL digital sphygmomanometer (Omron Healthcare Inc., Vernon Hills, IL, USA). The average of three blood pressure readings was recorded, with 5–10 min intervals between readings.

Fasting blood samples were obtained following a 10 h fast. Glucose and lipid profiles were measured on a Siemens Dimension RXL chemistry analyser (Diamond Diagnostics, Holliston, MA, USA). HbA1c was determined using a Variant device (Bio-Rad Laboratories, Hercules, GA, USA). All blood analysis was conducted at the clinical laboratories of Dasman Diabetes Institute. Research Electronic Data Capture was used for data collection and management.

### 2.3. Definition of Prediabetes and Diabetes

The WHO criteria for the diagnosis of diabetes was used: FPG ≥7 mmol/L or Hba1c ≥6.5% (48 mmol/mol) [8]. Participants were considered to have diabetes if they met one of the following: (1) self-reported a previous diagnosis of diabetes with concurrent prescribed glucose-lowering medication; (2) recorded a survey-measured FPG ≥7 mmol/L; or (3) recorded a survey-measured Hba1c ≥6.5%. Diabetes unawareness was determined as a survey measured elevated FPG or HbA1c without a prior self-reported diagnosis. Participants who self-reported a previous diabetes diagnosis, but were not currently receiving treatment for diabetes, and recorded a survey-measured FPG <7 mmol/L, and a survey-measured HbA1c <6.5%, were considered as “unverified” (*n* = 13). These individuals were recoded as having normal glycemia or prediabetes, if they met the prediabetes criteria.

For prediabetes, the WHO criteria for FPG was used [8] and the International Expert Committee criteria for HbA1c was used [9]. Participants were considered to have prediabetes if they did not meet any of the diabetes criteria and recorded a survey-measured FPG 6.1–6.9 mmol/L or a survey-measured HbA1c 6.0–6.4% (42–47 mmol/mol). Normal glycemia was defined as FPG ≤6 mmol/L and HbA1c ≤5.9% (≤41 mmol/mol) among individuals without diabetes.

### 2.4. Statistical Analysis

Only adults aged ≥20 years were included in the analysis; 10 participants aged 18 and 19 were excluded. As a result, a total of 4937 participants were included in the final analysis. Prevalence of diabetes and prediabetes was adjusted for age, sex and nationality to the 2011 Kuwait census recoded by the Public Authority for Civil Information [2]. Samples weights were calculated for Kuwaiti men, Kuwaiti women, non-Kuwaiti men and non-Kuwaiti women for four age groups: 20–29, 30–44, 45–59 and 60 and over. Samples weights for the 16 study strata were determined using the sample selection weight (population *n*/sample *n*), the nonresponse weight (1/response weight) and the population weight (population proportion/sample proportion), as previously described [4]. Non-Kuwaitis were divided into four subgroups: Arabs (predominately nationals of Egypt, Lebanon, Syria and Jordan), Iranians, South Asians (predominately nationals of India, Pakistan, Bangladesh and Sri Lanka) and Southeast Asians (predominately nationals of the Philippines).

Statistical significance was determined using the Student’s t test and the χ^2^ test. Normal weight was defined as a BMI <25 kg/m^2^, overweight was defined as a BMI 25.0–29.9 kg/m^2^ and obesity was defined as a BMI ≥30 kg/m^2^. Waist–hip ratios of ≥0.9 in men and ≥0.85 in women were considered elevated. Factors associated with diabetes and prediabetes were determined using a logistic regression model with prediabetes or diabetes as the binary dependent variable. Diabetes was not included as a dependent variable in the prediabetes model. The full model included sex, nationality, age, obesity status, waist–hip ratio, family history of diabetes, history of hypertension and dyslipidaemia, education level and monthly household income (*n* = 3441). Statistical analysis was performed using GraphPad Prism 8 (San Diego, CA, USA), IBM SPSS Statistics version 25 (Armonk, NY, USA) and R-Studio (Boston, MA, USA). A probability value ‘*p*’ < 0.05 was considered statistically significant.

### 2.5. Ethical Consideration

The study followed the Declaration of Helsinki ethical standards and was approved by the Ethical Review Committee at the Dasman Diabetes Institute. A written, informed consent was obtained from each participant prior to their inclusion in the study.

## 3. Results

The study participants’ demographics and clinical features are presented in Table 1. The proportion of male participants was 56.2% and the proportion of non-Kuwaiti participants was 65.9%. The overall mean age was 44.1 years and mean BMI was 30.1 kg/m^2^. The proportion of participants who self-reported a history of hypertension and dyslipidaemia was 20.9% and 22.6%, respectively. Family history of diabetes, defined as having at least one parent or sibling with diabetes, was reported in 61% of participants. Women had significantly higher mean BMI than men, as did Kuwaitis compared to non-Kuwaitis. Kuwaitis reported a higher frequency of self-reported hypertension, dyslipidaemia and family history of diabetes than non-Kuwaitis. Additionally, Kuwaitis reported higher educational levels and monthly household income.

The unadjusted prevalence of diabetes and prediabetes was 26.5% [95% CI, 25.3–27.7] and 14.9% [95% CI, 14.0–15.9], respectively. The adjusted prevalence of diabetes was 19.1% [95% CI, 18.0–20.2], and the adjusted prevalence of prediabetes was 13.5% [95% CI, 12.5–14.5] (Figure 1). When the HbA1c criteria for prediabetes was expanded from HbA1c 6–6.4% to HbA1c 5.7–6.4%, which is the diagnostic criteria recommended by the American Diabetes Association, adjusted prediabetes prevalence was 30.2% [95% CI, 28.9–31.5]. The adjusted prevalence of self-reported diabetes was 12.8% [95% CI, 11.9–13.7]; diabetes unawareness was 33.1% [95% CI 30.2–36.2].

The adjusted prevalence of diabetes in men was 22.4%, higher than the 14.4% found in women (Table 2). Likewise, the adjusted prevalence of prediabetes was higher in men (14.8%) than in women (11.5%). Diabetes prevalence was also higher in Kuwaitis (21.8%) than in non-Kuwaitis (18.2%). Amongst the non-Kuwaiti ethnicities, age and sex-adjusted diabetes prevalence was highest in South Asians (20.2%), followed by Arabs (19.1%), Iranians (15.2%) and finally, Southeast Asians (11.1%). Diabetes prevalence was significantly higher with age, from 5.4% in adults aged 20–29, to 14.2% in adults aged 30–44, 38.7% in adults aged 45–59 and 64.8% in adults aged 60 years or more. Prevalence of prediabetes and diabetes was higher with increased BMI and waist–hip ratio. A higher diabetes prevalence was also found in participants with self-reported hypertension, dyslipidaemia and a family history of diabetes. Individuals educated to a high-school level or lower reported a higher diabetes prevalence (24%) than in individuals with a university education (15.6%). In contrast, individuals with a monthly household income of 1000 Kuwaiti dinars or more (approximately $3300) reported a higher diabetes prevalence (20.9%) than those with a monthly household income of <1000 dinars (18.3%).

Kuwaiti men and Kuwaiti women had near identical diabetes prevalence (21.9% vs. 21.7%), and the prevalence of diabetes in Kuwaiti men (21.9%) was comparable to the 22.5% found in non-Kuwaiti men (Table 3). However, diabetes prevalence in Kuwaiti women was almost twice that in non-Kuwaiti women (21.7% vs. 11.0%). Diabetes prevalence was higher in Kuwaitis with self-reported hypertension and dyslipidaemia compared to non-Kuwaitis with those comorbidities, suggesting disease clustering is more common in Kuwaitis compared to non-Kuwaitis. Prediabetes was more prevalent in non-Kuwaitis than in Kuwaitis (Table 2 and Appendix A).

Table 4 shows the factors associated with prediabetes and diabetes. Older age, higher BMI and elevated waist–hip ratio were significantly associated with prediabetes. The risk of prediabetes in adults aged 45–59 and 60 years or more were double that in adults aged 20–29. Overweight adults had a prediabetes odds ratio of 1.41 and obese adults had an odds ratio of 1.6 when compared to normal-weight individuals. The risk of prediabetes in individuals with an elevated waist–hip ratio was 1.29 times than those with a normal waist–hip ratio.

The factors associated with prediabetes were also significantly associated with diabetes. The risk of diabetes in adults aged 45–59 was 2.91 times than in adults aged 20–29. Adults age 60 years or more had a diabetes odds ratio of 5. Obese adults were 1.82 times more prone to have diabetes than adults with a BMI < 25kg/m^2^, and an elevated waist–hip ratio increased the risk of diabetes by 2.45 times. Sex and medical history were also associated with diabetes. Risk of diabetes was significantly higher in men than women (OR = 1.78). Individuals who self-reported a previous diagnosis of hypertension and dyslipidaemia had a diabetes odds ratio of 2.29 and 2.58, respectively. Individuals with a family history of diabetes also had more than double the risk of diabetes (OR = 2.25). Finally, a lower education level was also associated with a higher risk of diabetes; adults educated to a high-school level or lower had an odds ratio of 1.53 relative to adults with a university education. Factors associated with diabetes and prediabetes in Kuwaiti nationals can be found in Appendix A, and factors associated with diabetes and prediabetes in non-Kuwaitis can be found in Appendix A. 

## 4. Discussion

In this nationally representative cross-sectional study, the estimated prevalence of diabetes was 19.1% and the estimated prevalence of prediabetes was 13.5%. Diabetes prevalence was higher in men than in women, and in Kuwaitis than in non-Kuwaitis. In addition, diabetes prevalence increased steeply with age; almost two-thirds of adults aged 60 years or more had diabetes and over three-quarters had either diabetes or prediabetes. Obesity, elevated waist–hip ratio and lower education level were all associated with diabetes. These results place Kuwait amongst the countries with the highest diabetes prevalence in the world.

A few other previous studies have also reported on diabetes prevalence in Kuwait. A 2006 WHO STEPS cross-sectional study of 1970 Kuwaiti adults aged 20–65 reported a diabetes prevalence of 17.9% [10]. A subsequent WHO STEPS study in 2014 of 2561 Kuwaiti adults aged 18–69 reported an estimated adjusted diabetes prevalence of 18.8% [4]. Whereas this study sampled all adults in Kuwait, both the 2006 and 2014 WHO STEPS studies sampled only Kuwaiti citizens, which make up approximately 30% of the country’s population. Kuwait’s large, multiethnic expatriate majority was excluded. The estimated adjusted prevalence of diabetes among Kuwaiti adults in this study was 21.8%, higher than the reported figures from both WHO STEPS studies.

We found that diabetes prevalence among non-Kuwaitis was 18.2%. Although high, this was lower than the prevalence among Kuwaitis. While non-Kuwaiti men had a similar diabetes prevalence to Kuwaiti men, non-Kuwaiti women had half the diabetes prevalence compared to Kuwaiti women. Amongst non-Kuwaiti Arabs, which are the largest expatriate community in Kuwait, diabetes prevalence was 19.1%, which was slightly higher than diabetes prevalence reported in Egypt (16.8%) [11], Syria (15.6%) [12], Lebanon (18.0%) [13] and Jordan (17.1%) [14]. However, the 20.2% diabetes prevalence estimated here in Kuwait’s South Asian expatriate community was much higher than the reported diabetes prevalence in India (7.7%) [15], Sri Lanka (10.3%) [16] and Bangladesh (9.7%) [17], although a recent study estimated that diabetes prevalence could be as high as 17% in Pakistan [18]. Prevalence of diabetes in the South Asian communities of the United Arab Emirates [19], United Kingdom [20] and United States [21] were also reportedly higher than in their native lands. We also found higher diabetes prevalence amongst Iranians and Southeast Asians (the majority of which were Filipino) than the estimated diabetes prevalence in Iran (11.4%) [22] and the Philippines (7.2%) [23].

To our knowledge, the only other national survey to sample both citizens and expatriates in Kuwait was the World Health Survey Plus (WHS+). The WHS+ was conducted between 2008 and 2009 in Kuwait and other countries in the region and surveyed 2518 citizens and 1310 expatriates aged 18 years or older in Kuwait [24]. However, the WHS+ survey only measured self-reported diabetes. Adjusted self-reported diabetes in Kuwait was estimated in the WHS+ to be 15.3%, higher than the 12.8% found in this study. Kuwaitis were reported to have almost double the prevalence of self-reported diabetes compared to expatriates in the WHS+ (14% vs. 7.2%).

The estimated adjusted prediabetes prevalence in this study was 13.5%, defined as a study-recorded FPG of 6.1–6.9 mmol/L or an HbA1c of 6.0–6.4% in individuals without treated diabetes. The estimated prediabetes prevalence in the 2014 WHO STEPS study, defined as a study-recorded FPG of 6.1–6.9 mmol/L or an HbA1c of 5.7–6.4% in individuals without treated diabetes, was 19.4%. Using this expanded definition, the estimated adjusted prediabetes prevalence in this study was 30.2%, which, although high, was lower than the reported 37.5% crude prediabetes prevalence in the United States [25]. There is currently a lack of consensus on how to diagnose prediabetes, with the WHO [8], the IEC [9] and the ADA [26] using different HbA1c criteria. Using different diagnostic criteria can have a large effect on prevalence estimates [27].

Diabetes prevalence is high across the Middle East. A recent meta-analysis of 50 studies in 15 Middle Eastern countries with a pooled population of 4.3 million people estimated the combined prevalence of diabetes in the region was 14.6% [28]. Israel had the lowest reported diabetes prevalence in the region (2.6%), whereas the countries of the Gulf Cooperation Council had the highest reported prevalence: 16.5% in Qatar, 21.4% in Saudi Arabia and 21.6% in the United Arab Emirates. The meta-analysis also found a steady increase in diabetes prevalence with time. Diabetes prevalence increased from 8.4% in studies from 2000 to 2005 to 18.9% in studies from 2010 to 2017 [28]. In Kuwait, the 19.1% diabetes prevalence reported in this study is a 2.5-fold increase from the 7.6% crude diabetes prevalence reported in 1996 [29].

The association between obesity and diabetes is well established [30] and obesity is the major contributing factor to the high diabetes prevalence in Kuwait. Elevated BMI and waist–hip ratio were associated with both prediabetes and diabetes in this study. Obesity in Kuwait ranks amongst the highest in the world. Over 40% of Kuwaiti adults were obese and 37% were overweight in 2014 [3]. We also found an inverse correlation between education level and diabetes; adults without a university education had 50% higher risk of developing diabetes than those with a university education, consistent with findings from developed countries reportedly showing an inverse association between education level and diabetes, overall health and mortality [31,32,33]. In contrast, we found that diabetes prevalence increased with income. The risk of diabetes is generally associated with lower income [34], although a recent study reported higher diabetes risk with increased household wealth in low-income and low-to-middle-income countries [35]. There is a large disparity in income between Kuwaiti nationals and expatriates.

The staggering increase in diabetes prevalence with age is alarming. Almost two-thirds of adults aged 60 years or above had diabetes in this study, over three times higher than the global estimate for adults over 60 years. In fact, the estimated diabetes prevalence reported here outpaces the estimated global prevalence at every age group [36]. Estimated prediabetes prevalence peaked here at the 45–59 age group before decreasing, suggesting the transition of a growing number of individuals with prediabetes to diabetes with age. The diabetes pandemic in Kuwait is not limited to adults. Type 2 diabetes prevalence in Kuwaiti children aged 6–18 years was estimated to be 35 per 100,000 [37], and the childhood obesity prevalence is over 30% [38]. As Kuwait has a relatively young population [2], the socioeconomic burden of diabetes will likely increase as adolescents and young adults grow older should this trend continue.

Our study strengths include the large sample size and the nationally representative study population. KDEP was the first cross-sectional survey to assess HbA1c and FPG levels amongst both nationals and expatriates in Kuwait. However, KDEP did not include a 2 h oral glucose tolerance test (2hrOGTT), often considered the most reliable and sensitive method for diagnosing diabetes [39]. As a result, the diabetes prevalence estimates reported here may be underestimates. A large international pooled analysis of health surveys reported that using FPG or 2hrOGTT captures the largest number of people with diabetes, although they also reported geographical differences [27]. A study in the United States reported that almost half of individuals with undiagnosed diabetes may be missed in the absence of a 2hrOGTT diagnosis [40]. KDEP was conducted between 2011 and 2014; sample collection ended six years ago. Given the recent upward trend, it is possible that the prevalence of diabetes and prediabetes in Kuwait in 2020 is higher than reported here. As KDEP was a cross-sectional survey, we could not assess temporal relationships between diabetes prevalence and risk factors. Another limitation of this study was that a distinction between type 1 and type 2 diabetes was not determined in this survey. Although the incidence of childhood-onset type 1 diabetes in Kuwaiti nationals is 41 per 100,000 per year, one of the highest in the world [41], the majority of people with diabetes in Kuwait have type 2 diabetes.

In conclusion, Kuwait has one of the largest diabetes burdens in the world. The high rates of diabetes, obesity and associated comorbidities across all segments of society impose a substantial burden on Kuwait’s health system. A sugar excise tax on sodas and energy drinks was introduced by the Gulf Cooperation Council in 2017, and Kuwait was set to implement the tax in 2020 [42]. Public health interventions have led to a stabilisation and, in some countries, a fall in diabetes incidence over the past decade [43]. Kuwait needs to follow suit if it is to manage its growing diabetes pandemic.

## Figures and Tables

**Figure 1 jcm-09-03420-f001:**
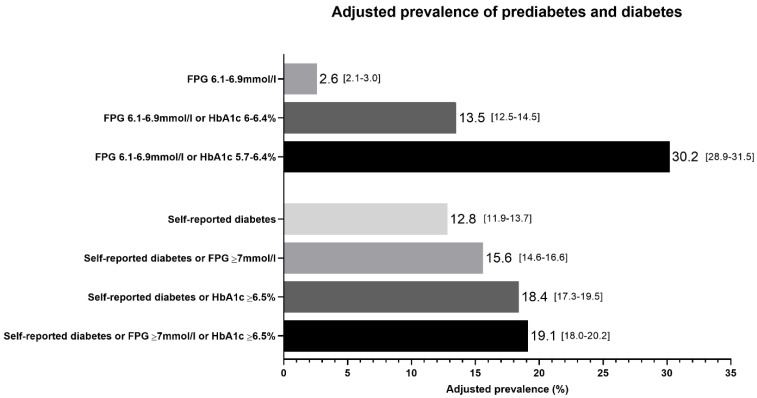
The adjusted prevalence of diabetes and prediabetes according to different diagnostic criteria (*n* = 4937). KDEP Survey, adults in Kuwait aged 20 years or more. Prevalence adjusted for age, sex and nationality and presented with 95% confidence intervals.

**Table 1 jcm-09-03420-t001:** Participants’ demographic and clinical characteristics according to sex and nationality ^a^. KDEP Survey, adults in Kuwait aged 20 years or more.

		Sex			Nationality		
Characteristic	Total	Male	Female	*p*	Kuwaiti	Non-Kuwaiti	*p*
*n* (% total)	4937 (100)	2773 (56.2)	2164 (43.8)		1684 (34.1)	3253 (65.9)	
Mean age (years)	44.1 (10.4)	44.6 (10.4)	43.4 (10.3)	<0.0001	46.7 (10.7)	42.7 (9.9)	<0.0001
Mean BMI (kg/m^2^)	30.1 (5.8)	29.5 (5.28)	30.9 (6.41)	<0.0001	31.2 (5.7)	29.6 (5.8)	<0.0001
Mean HbA1c (%)	6.16 (1.5)	6.28 (1.56)	6.0 (1.4)	<0.0001	6.27 (1.54)	6.1 (1.47)	0.0001
Mean FPG (mmol/L)	5.95 (2.29)	6.17 (2.39)	5.67 (2.12)	<0.0001	6.45 (2.4)	5.84 (2.22)	<0.0001
Mean systolic BP (mmHg)	130 (21.1)	135 (20.9)	125 (20)	<0.0001	126 (18.8)	132 (22)	<0.0001
Mean diastolic BP (mmHg)	79 (12.1)	81 (12.0)	77 (11.9)	<0.0001	77 (11.2)	81 (12.3)	<0.0001
Mean waist–hip ratio (cm/cm)	0.91 (0.07)	0.94 (0.06)	0.88 (0.07)	<0.0001	0.92 (0.06)	0.91 (0.07)	0.61
Self-reported hypertension (*n*)	1002 (20.9)	593 (22.1)	409 (19.3)	0.02	380 (22.6)	622 (19.9)	0.027
Self-reported dyslipidaemia (*n*)	1046 (22.6)	624 (24.1)	422 (20.6)	0.004	409 (24.5)	637 (21.5)	0.02
Family history of diabetes (*n*)	2807 (61.0)	1567 (60.1)	1240 (62.2)	0.16	1178 (78.1)	1629 (52.7)	<0.0001
University educated (*n*)	2807 (56.9)	1549 (55.9)	1258 (58.1)	0.11	1166 (69.2)	1641 (50.5)	<0.0001
>1000 KD monthly household income (*n*)	1778 (44.4)	1001 (42.7)	777 (46.9)	0.008	1321 (92.4)	457 (17.8)	<0.0001

^a^ Data represents mean with standard deviation in parenthesis or absolute numbers (*n*) with percentage in parenthesis. *p* values for categorical variables were obtained by χ^2^ test. *p* values for continuous variables were obtained by the Student’s t test. KD = Kuwaiti Dinar. 1000 KD is approximately US $3300.

**Table 2 jcm-09-03420-t002:** Adjusted prevalence of prediabetes and diabetes in adults aged 20 years or more.

		Prediabetes Prevalence		Diabetes Prevalence	
Characteristic	*n*	% [95% CI]	*p*	% [95% CI]	*p*
**Overall**	4937	13.5 [12.5–14.5]		19.1 [18.0–20.2]	
**Sex**			0.0008		<0.0001
Men	2773	14.8 [13.5–16.2]		22.4 [20.9–24.0]	
Women	2164	11.5 [10.2–12.9]		14.4 [13.0–15.9]	
**Nationality**			0.002		0.003
Kuwaiti	1684	11.1 [9.7–12.7]		21.8 [19.9–23.8]	
Non-Kuwaiti	3253	14.3 [13.1–15.5]		18.2 [16.9–19.6]	
**Non-Kuwaiti ethnicities**			0.005		0.0004
Arab	1615	12.5 [11.0–14.2]		19.1 [17.3–21.1]	
Iranian	150	11.3 [7.1–17.3]		15.2 [10.4–21.8]	
South Asian	1051	17.2 [15.0–19.6]		20.2 [17.8–22.7]	
Southeast Asian	407	14.7 [11.6–18.5]		11.1 [8.4–14.5]	
**Age group**			0.001		<0.0001
20–29	370	8.4 [6.0–11.6]		5.4 [3.5–8.2]	
30–44	2282	14.3 [12.9–15.8]		14.2 [12.9–15.7]	
45–59	1889	18.6 [16.9–20.4]		38.7 [36.5–40.9]	
60+	396	12.2 [9.3–15.7]		64.8 [60.0–69.4]	
**Obesity category**			0.002		<0.0001
Normal BMI	825	9.7 [7.9–11.9]		8.4 [6.7–10.5]	
Overweight	1888	14.4 [12.8–16.0]		17.6 [16.0–19.4]	
Obese	2219	14.8 [13.4–16.3]		26.7 [24.9–28.6]	
**Waist–hip ratio**			<0.0001		<0.0001
Normal	1231	9.1 [7.6–10.8]		7.1 [5.8–8.6]	
Elevated	3664	15.7 [14.6–16.9]		25.1 [23.7–26.5]	
**Self-reported hypertension**			0.63		<0.0001
No	3800	13.3 [12.2–14.4]		14.1 [13.1–15.3]	
Yes	1002	13.9 [11.9–16.2]		51.5 [48.4–54.6]	
**Self-reported dyslipidaemia**			0.088		<0.0001
No	3591	13.1 [12.1–14.3]		13.9 [12.8–15.0]	
Yes	1046	15.2 [13.2–17.5]		47.8 [44.8–50.8]	
**Family history of diabetes**			0.62		<0.0001
No	1795	13.3 [11.8–15.0]		11.9 [10.5–13.4]	
Yes	2807	13.8 [12.6–15.1]		24.5 [23.0–26.1]	
**Education level**			0.92		<0.0001
High school or lower	2129	13.5 [12.1–15.0]		24.0 [22.2–25.8]	
University	2807	13.4 [12.2–14.7]		15.6 [14.3–17.0]	
**Monthly household income**			0.049		0.042
≤1000 KD (≤approx. $3300)	2223	14.2 [12.8–15.7]		18.3 [16.7–20.0]	
>1000 KD (>approx. $3300)	1778	12.1 [10.7–13.7]		20.9 [19.0–22.8]	

**Table 3 jcm-09-03420-t003:** Adjusted prevalence of diabetes by nationality in adults aged 20 years or more.

	Kuwaitis		Non-Kuwaitis		
	*n*	Prevalence, % [95% CI]	*n*	Prevalence, % [95% CI]	*p*
**All**	1684	21.8 [19.9–23.8]	3253	18.2 [16.9–19.6]	0.0025
**Sex**					
Men	863	21.9 [19.3–24.8]	1910	22.5 [20.7–24.4]	0.74
Women	821	21.7 [19.0–24.6]	1343	11.0 [9.5–12.8]	<0.0001
**Age**					
20–29	78	6.3 [2.7–14.0]	292	5.1 [3.1–8.3]	0.66
30–44	679	12.7 [10.4–15.4]	1603	14.6 [13.0–16.4]	0.22
45–59	724	39.0 [35.5–42.6]	1165	38.6 [35.9–41.5]	0.89
60+	203	69.5 [62.8–75.4]	193	59.3 [52.3–65.9]	0.04
**Obesity category**					
Normal BMI	173	8.1 [4.9–13.1]	652	8.4 [6.5–10.8]	0.88
Overweight	600	17.2 [14.4–20.4]	1288	17.8 [15.8–20.0]	0.75
Obese	909	29.9 [27.0–33.0]	1310	25.3 [23.0–27.7]	0.015
**Waist–hip ratio**					
Normal	338	8.6 [6.0–12.0]	893	6.7 [5.3–8.6]	0.26
Elevated	1326	28.4 [26.0–30.8]	2338	23.9 [22.2–25.7]	0.003
**Self-reported hypertension**					
No	1299	13.9 [12.2–15.9]	2501	14.2 [12.9–15.6]	0.83
Yes	380	68.2 [63.4–72.7]	622	45.4 [41.6–49.4]	<0.0001
**Self-reported dyslipidaemia**					
No	1263	12.7 [10.9–14.6]	2328	14.3 [13.0–15.8]	0.16
Yes	409	67.7 [63.0–72.1]	637	40.3 [36.5–44.1]	<0.0001
**Family history of diabetes**					
No	331	14.2 [10.8–18.4]	1464	11.5 [9.9–13.2]	0.17
Yes	1178	23.2 [20.9–25.7]	1629	25.2 [23.1–27.3]	0.22
**Education level**					
High school or lower	518	35.5 [31.5–39.7]	1611	21.7 [19.8–23.8]	<0.0001
University	1166	16.7 [14.7–19.0]	1641	15.1 [13.4–16.9]	0.23
**Monthly household income**					
≤1000 KD (≤approx. $3300)	109	19.3 [13.0–27.7]	2114	18.3 [16.7–20.0]	0.79
>1000 KD (>approx. $3300)	1321	22.0 [19.9–24.3]	457	18.6 [15.3–22.4]	0.12

**Table 4 jcm-09-03420-t004:** Factors associated with prediabetes and diabetes in adults aged 20 years or more.

	Prediabetes		Diabetes	
Characteristic	OR [95% CI]	*p*	OR [95% CI]	*p*
**Sex**				
Women	1.00		1.00	
Men	1.11 [0.91–1.36]	0.29	1.78 [1.47–2.16]	<0.001
**Nationality**				
Non-Kuwaiti	1.00		1.00	
Kuwaiti	0.84 [0.63–1.12]	0.23	1.10 [0.84–1.44]	0.48
**Age group**				
20–29	1.00		1.00	
30–44	1.47 [0.93–2.33]	0.10	1.08 [0.63–1.85]	0.78
45–59	1.97 [1.23–3.15]	0.005	2.91 [1.70–4.97]	<0.001
60+	2.03 [1.14–3.62]	0.016	5.00 [2.75–9.10]	<0.001
**Obesity category**				
Normal BMI	1.00		1.00	
Overweight	1.41 [1.03–1.92]	0.031	1.23 [0.90–1.67]	0.19
Obese	1.60 [1.17–2.19]	0.003	1.82 [1.35–2.47]	<0.001
**Waist–hip ratio**				
Normal	1.00		1.00	
Elevated	1.29 [1.00–1.65]	0.044	2.45 [1.87–3.22]	<0.001
**Self-reported hypertension**				
No	1.00		1.00	
Yes	0.79 [0.60–1.03]	0.079	2.29 [1.85–2.83]	<0.001
**Self-reported dyslipidaemia**				
No	1.00		1.00	
Yes	0.91 [0.71–1.17]	0.48	2.58 [2.11–3.16]	<0.001
**Family history of diabetes**				
No	1.00		1.00	
Yes	0.93 [0.76–1.14]	0.47	2.25 [1.83–2.76]	<0.001
**Education level**				
University	1.00		1.00	
High school or lower	0.79 [0.64–0.97]	0.022	1.53 [1.27–1.85]	<0.001
**Monthly household income**				
>1000 KD (>approx. $3300)	1.00		1.00	
≤1000 KD (≤approx. $3300)	1.24 [0.93–1.63]	0.14	1.30 [0.99–1.69]	0.058

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
