# Peer review of "Adult Diabetes and Prediabetes Prevalence in Kuwait: Data from the Cross-Sectional Kuwait Diabetes Epidemiology Program"

_jcm, 2020, doi:10.3390/jcm9113420_

Round 1
Reviewer 1 Report
Nice and very usefull work giving data for diabetes and prediabetes. I thnk in a future work the authors in a part of their population have to a)OGTT b) documente the physical exercise frecuency
Author Response
We thank the reviewer for their kind comments. We agree for the need for future studies that include OGTT and measures of physical exercise.
Reviewer 2 Report
- In view of the marked differences in characteristics for Kuwaiti citizens and non-citizens, I suggest that the authors break down the display and analysis of results in Tables 2 and 3 and Figure 1, to show the non-Kuwaitis separately from the Kuwaitis. I regard it as optional for you to show the results for the combined populations, as currently shown in Fig. 1 and Tables 1 and 2 in addition to the breakdown into the two groups. (this could be put into supplementary data, if the journal allows that kind of feature).
- It might be good to show the percentage of non-Kuwaitis from various countries throughout the world, at least for perhaps the largest categories.
- Would be good to show trend analysis - how they stand now relatively to 5, 10, or more years ago.
- Might be good to compare with the results from other countries in close geographical proximity to the extent that such data are currently available in the literature or from IDF.
- Include information on previous programs applied in Kuwait or in surrounding or neighboring countries to try to reduce the incidence and prevalence of prediabetes and diabetes.
- Discuss type 1 diabetes in Kuwait at least briefly to put that into perspective.
- Discuss the current cost to the health care system to the extent it is available.
- Discus prevalence and costs of complications of diabetes, a) currently, in current year, and b) projected out 5 and 10 years, if the prevalence and incidence continue to grow at rates based on data from the last 10 years.
- What are the relative costs for Kuwait's and non-Kuwaitis, now and projected to 5 and 10 years.
- As non-Kuwaiti's reside and work in Kuwait for increasing lengths of time, what happens to their risk for prediabetes and for diabetes? Do they become more like the Kuwaiti's in that regard, due to increasing affluence, change in diet or other customs. Compare non-Kuwaitis after 1, 5 10 years of residence in Kuwait. Does this vary by income, sex, or working/non-working status?
Author Response
1. In view of the marked differences in characteristics for Kuwaiti citizens and non-citizens, I suggest that the authors break down the display and analysis of results in Tables 2 and 3 and Figure 1, to show the non-Kuwaitis separately from the Kuwaitis. I regard it as optional for you to show the results for the combined populations, as currently shown in Fig. 1 and Tables 1 and 2 in addition to the breakdown into the two groups. (this could be put into supplementary data, if the journal allows that kind of feature).
Response: Thank you for your helpful and considerate comments. We have added a new table to the manuscript (a new Table 3) which breaks down diabetes prevalence by nationality (Kuwaiti v Non-Kuwaitis). We’ve also added 3 new supplementary tables to the Appendix; one tabulates pre-diabetes prevalence by nationality, one tabulates factors associated with diabetes and pre-diabetes in Kuwaitis and the third tabulates factors associated with diabetes and pre-diabetes in non-Kuwaitis. We’ve also added a few lines to the Results and Discussion to reflect the new additions. Results; lines 162-168 and 185-187. Discussion; lines 212-214.
We’ve opted to keep the results showing prevalence in the entire population as we feel that’s of great interest to the readership.
2. It might be good to show the percentage of non-Kuwaitis from various countries throughout the world, at least for perhaps the largest categories.
Response: Non-Kuwaitis were divided into 4 subgroups reflecting the largest expatriate populations in the country; Arabs (predominately nationals of Egypt, Lebanon, Syria and Jordan), Iranians, South Asians (predominately nationals of India, Pakistan, Bangladesh and Sri Lanka) and South East Asians (predominately nationals of the Philippines). The manuscript has been amended to include this information in the Methods, lines 112-115.
3. Would be good to show trend analysis - how they stand now relatively to 5, 10, or more years ago.
Response: Unfortunately, KDEP was a cross-sectional study and as such a trend analysis was not possible. The diabetes prevalence reported in this study is a 2.5 fold increase from the 7.6% crude diabetes prevalence found in a 1996 study. The discussion has been amended to mention the increasing trend of diabetes prevalence in Kuwait and the wider Middle East. Discussion; lines 246-249.
4. Might be good to compare with the results from other countries in close geographical proximity to the extent that such data are currently available in the literature or from IDF.
Response: The discussion has been amended to include a paragraph discussing diabetes prevalence across the Middle East. Discussion; lines 241-246.
5. Include information on previous programs applied in Kuwait or in surrounding or neighboring countries to try to reduce the incidence and prevalence of prediabetes and diabetes.
Response: Unfortunately, comprehensive diabetes prevention programs haven’t been implemented yet in Kuwait or the surrounding countries of the Persian Gulf. However, as mentioned in the last paragraph of the discussion, a sugar excise tax on sodas and energy drinks was introduced by the Gulf Cooperation Council in 2017, which is an important first step. Kuwait was set to implement the tax in 2020 but due to the Covid-19 global pandemic this has been delayed.
6. Discuss type 1 diabetes in Kuwait at least briefly to put that into perspective.
Response: A distinction between type 1 and type 2 diabetes was not recorded in this study, which was unfortunate because Kuwait has one of the highest incidences of childhood type 1 diabetes in the world. The manuscript has been amended to include a mention of this in the discussion. Discussion; lines 284-287.
7. Discuss the current cost to the health care system to the extent it is available.
8. Discus prevalence and costs of complications of diabetes, a) currently, in current year, and b) projected out 5 and 10 years, if the prevalence and incidence continue to grow at rates based on data from the last 10 years.
9. What are the relative costs for Kuwait's and non-Kuwaitis, now and projected to 5 and 10 years.
Response to 7-9: There hasn’t been a comprehensive study on the socioeconomic impact of diabetes in Kuwait. As mentioned in the introduction, the IDF estimates that Kuwait spends over $1,000 per adult with diabetes. Unfortunately, that’s the extent of our knowledge regarding the financial burden of diabetes in the country. The cost of diabetes on society and the economy in Kuwait is an important question and one that’ll hopefully be addressed in a future study soon. However, that was beyond the scope of the this study.
10. As non-Kuwaiti's reside and work in Kuwait for increasing lengths of time, what happens to their risk for prediabetes and for diabetes? Do they become more like the Kuwaiti's in that regard, due to increasing affluence, change in diet or other customs. Compare non-Kuwaitis after 1, 5 10 years of residence in Kuwait. Does this vary by income, sex, or working/non-working status?
Response: KDEP was not a longitudinal study so a trend analysis wasn’t possible. However, as mentioned in the discussion, the prevalence of diabetes in the South Asian community in Kuwait is comparable to the prevalence in Kuwaiti nationals and is much higher than the recorded prevalence in India, Sri Lanka and Bangladesh. This finding is not unique to Kuwait – South Asian expatriates in the UAE, USA and UK have higher diabetes rates than the prevalence in their native lands.
In Kuwait, as can be seen in the new Table 3 of this study, diabetes prevalence is high in both low and high income non-Kuwaiti expatriates in the country although there is a gender gap – diabetes prevalence in non-Kuwaiti men is twice that found in non-Kuwaiti women. Many more female than male expatriates in Kuwait come from South East Asian countries where the occurrence of diabetes is lower than in South Asian and Arab countries from where more males come.
Reviewer 3 Report
This study was aimed to estimate the prevalence of diabetes in adults residing in Kuwait. The authors conducted a population-based cross-sectional survey of non-communicable diseases using a representative stratified random sample of all residents aged >= 18 years in Kuwait, under the Kuwait Diabetes Epidemiology Program. The study period was 2011-2014, the WHO STEPwise approach was used to collect the data and the participation rate was equal 85.1%. The same methodology had previously been used in a study with the same objectives but based on the adult population aged 18-69 years and including only Kuwaiti citizens. Overall, 5291 people completed the interviews and 4947 adults completed the interviews and provided a biochemical analysis of the blood.
Minors
- Since it is a cross-sectional study, it is better to deal with factors associated with diabetes or pre-diabetes, rather than risk factors (lines 114,167, 163, table 3]. Please change throughout the manuscript.
- Please use p in small caps to indicate p-value (lines 120, 175, table 1, table 2 and 3)
- In Table 1, put standard deviations in brackets: mean (SD), rather than mean ± SD
- Please, check in table 3 confidence interval and p-value for the ODDS ratio of educational status in pre-diabetes condition: the former does not contain the unit, so p-value should be less than 0.05.
Majors
The discussion should be improved by focusing on the results obtained by investigating factors associated with the two conditions analysed.
The discussion should include limitations if the study, highlighting that the findings refer to more than six years ago.
Author Response
This study was aimed to estimate the prevalence of diabetes in adults residing in Kuwait. The authors conducted a population-based cross-sectional survey of non-communicable diseases using a representative stratified random sample of all residents aged >= 18 years in Kuwait, under the Kuwait Diabetes Epidemiology Program. The study period was 2011-2014, the WHO STEPwise approach was used to collect the data and the participation rate was equal 85.1%. The same methodology had previously been used in a study with the same objectives but based on the adult population aged 18-69 years and including only Kuwaiti citizens. Overall, 5291 people completed the interviews and 4947 adults completed the interviews and provided a biochemical analysis of the blood.
Minors
1. Since it is a cross-sectional study, it is better to deal with factors associated with diabetes or pre-diabetes, rather than risk factors (lines 114,167, 163, table 3]. Please change throughout the manuscript.
Response: Amended as requested.
2. Please use p in small caps to indicate p-value (lines 120, 175, table 1, table 2 and 3)
Response: Amended as requested.
3. In Table 1, put standard deviations in brackets: mean (SD), rather than mean ± SD
Response: Amended as requested.
4. Please, check in table 3 confidence interval and p-value for the ODDS ratio of educational status in pre-diabetes condition: the former does not contain the unit, so p-value should be less than 0.05.
Response: Thank you for spotting this typographical error. The p-value was 0.022. This has been amended.
Majors
The discussion should be improved by focusing on the results obtained by investigating factors associated with the two conditions analysed.
Response: The discussion has been amended to include a paragraph discussing the association of obesity and education status with diabetes. Discussion; lines 250-261.
The discussion should include limitations if the study, highlighting that the findings refer to more than six years ago.
Response: The penultimate paragraph in the discussion highlighting the limitations of the study has been amended to include a mention that the study sample collection ended in 2014. Discussion; lines 280-282.
Reviewer 4 Report
This paper is very well and clearly written and needs in my opinion. Readers will learn a lot.
Author Response
We thank the reviewer for their kind comments.
Round 2
Reviewer 3 Report
Authors have amended the manuscript as requested. The manuscript itself is considerably improved
I am sorry to add a small detail, p-values in tables should have the same numbers of decimals, and three are generally sufficient. So, 0.67 and 0.0001 become 0.670 and <0.001.